# Statins Show Anti-Atherosclerotic Effects by Improving Endothelial Cell Function in a Kawasaki Disease-like Vasculitis Mouse Model

**DOI:** 10.3390/ijms232416108

**Published:** 2022-12-17

**Authors:** Yusuke Motoji, Ryuji Fukazawa, Ryosuke Matsui, Yoshinori Abe, Ikuno Uehara, Makoto Watanabe, Yoshiaki Hashimoto, Yasuo Miyagi, Noriko Nagi-Miura, Nobuyuki Tanaka, Yosuke Ishii

**Affiliations:** 1Department of Cardiovascular Surgery, Nippon Medical School Hospital, 1-1-5 Sendagi, Bunkyo-ku, Tokyo 113-8603, Japan; 2Department of Pediatrics, Nippon Medical School Hospital, 1-1-5 Sendagi, Bunkyo-ku, Tokyo 113-8603, Japan; 3Department of Molecular Oncology, Institute for Advanced Medical Sciences, Nippon Medical School, 1-1-5 Sendagi, Bunkyo-ku, Tokyo 113-8603, Japan; 4Laboratory for Immunopharmacology of Microbial Products, Tokyo University of Pharmacy and Life Sciences, Hachioji 192-0392, Japan

**Keywords:** Kawasaki disease, statin, *Candida albicans* water-soluble, vasculitis, eNOS, atherosclerosis, cellular senescence, Akt

## Abstract

Kawasaki disease (KD) is an acute inflammatory syndrome of unknown etiology that is complicated by cardiovascular sequelae. Chronic inflammation (vasculitis) due to KD might cause vascular cellular senescence and vascular endothelial cell damage, and is a potential cause of atherosclerosis in young adults. This study examined the effect of KD and HMG-CoA inhibitors (statins) on vascular cellular senescence and vascular endothelial cells. *Candida albicans* water-soluble fraction (CAWS) was administered intraperitoneally to 5-week-old male apolipoprotein E-deficient (ApoE−) mice to induce KD-like vasculitis. The mice were then divided into three groups: control, CAWS, and CAWS+statin groups. Ten weeks after injection, the mice were sacrificed and whole aortic tissue specimens were collected. Endothelial nitric oxide synthase (eNOS) expression in the ascending aortic intima epithelium was evaluated using immunostaining. In addition, eNOS expression and levels of cellular senescence markers were measured in RNA and proteins extracted from whole aortic tissue. KD-like vasculitis impaired vascular endothelial cells that produce eNOS, which maintains vascular homeostasis, and promoted macrophage infiltration into the tissue. Statins also restored vascular endothelial cell function by promoting eNOS expression. Statins may be used to prevent secondary cardiovascular events during the chronic phase of KD.

## 1. Introduction

Kawasaki disease (KD) is an acute inflammatory syndrome of unknown etiology that develops from severe coronary arteritis and coronary artery lesions (CAL) such as coronary artery aneurysms, stenoses, and occlusions, and potentially results in death [1,2].

Long-term observation of patients with KD has resulted in reports of CAL developing in childhood and acute coronary syndromes developing in early adulthood, even in cases where CAL have morphologically regressed following treatment of the acute disease [2].

KD can promote atherosclerosis, especially in cases of CAL formation where vascular remodeling is observed [2]. Therefore, life-long follow-up is necessary. Pathologically, KD coronary aneurysm lesions are predominantly observed during post-inflammatory sclerosis, and have a different histology from that of typical atherosclerosis [3,4,5]. However, vascular cellular senescence and chronic vascular endothelial dysfunction caused by chronic inflammation may contribute to the pathogenesis of atherosclerosis in early adulthood [3,4,5].

We previously reported prolonged endothelial cell dysfunction and vascular senescence in KD aneurysm specimens from young patients that were similar to those observed in adults with atherosclerotic lesions [6]. There is currently no direct evidence of KD and its potential for progression to atherosclerosis in early adulthood, and few basic experimental studies have been reported. Using an animal model of KD-like vasculitis caused by *Lactobacillus casei* cell wall extract (LCWE), Arditi et al. reported that complications from dyslipidemia—a risk factor for atherosclerosis—and KD could increase the risk of early onset atherosclerosis [7]. We also previously established a mouse model of KD-like vasculitis using the *Candida albicans* water-soluble fraction (CAWS), and reported that KD-like vasculitis not only infiltrates inflammatory cells (mainly macrophages) in the vascular tissue to form CAL and aortic aneurysms during the later stage, but also promotes systemic atherosclerosis [8]. This was the first animal study to demonstrate that an HMG-CoA inhibitor (statin) has a therapeutic effect on persistent vasculitis and atherosclerosis [8].

However, although statins suppress the infiltration of tissues by inflammatory cells resulting from KD-like vasculitis, the underlying mechanism has not been investigated. Vascular endothelial cells, which organize the systemic vascular intima, have an important role in maintaining homeostasis, and protecting tissues from inflammatory cell invasion [9,10,11].

Endothelial nitric oxide synthase (eNOS), which is expressed specifically in vascular endothelial cells, is primarily associated with the production of the vasoactive substance nitric oxide (NO), and is an important factor when assessing endothelial cell function [12,13,14,15]. Previous studies have shown that statins are associated with the expression of eNOS mediated by Akt [16,17,18]. CAWS vasculitis has also been speculated to induce cellular senescence of vascular endothelial cells, leading to suppression of eNOS [19]. When assessing cellular senescence, it is important to use multiple markers of cellular senescence such as phospho-retinoblastoma protein (phospho-Rb), and intercellular adhesion molecule 1 (*ICAM-1*) [20,21].

In this study, we generated a KD-like vasculitis model using the atherosclerotic animal model, apolipoprotein-E-deficient (ApoE−) mice, and examined how CAWS vasculitis and statins affect vascular senescence and endothelial cell function. We also examined the mechanisms underlying these effects.

## 2. Results

### 2.1. Immunohistochemical Staining of Frozen Sections of Ascending Aortic Tunica Intima

Continuous eNOS expression was observed in the ascending aortic tunica intima of mice in the control group, while mice in the CAWS group had very low eNOS expression. Partially continuous eNOS expression was observed in the intima of mice in the CAWS+statin group, although this was lower than that observed in mice in the control group (Figure 1a).

### 2.2. Scoring the Intensity of eNOS Expression in Ascending Aortic Tunica Intima

The intensity of eNOS expression was quantified, as described in the Methods section. eNOS expression was significantly lower in the CAWS group compared with the control group (25 [27.7, 21] vs. 2.3 [9.2, 1.7], [*p* = 0.012]; Table 1 and Figure 1b), but was significantly higher in the CAWS+statin group compared with the CAWS group (2.3 [9.2, 1.7] vs. 20 [22.5, 17.5], [*p* = 0.012]; Table 1 and Figure 1b). eNOS expression in the CAWS+statin group was lower than in the control group, although the difference was not statistically significant (25 [27.7, 21] vs. 20 [22.5, 17.5], [*p* = 0.060]; Table 1 and Figure 1b).

### 2.3. Expression of eNOS Protein in Whole Aortic Tissue

The expression of eNOS protein was significantly higher in the control group than in the CAWS group (3.5 ± 1.1 vs. 0.9 ± 0.2 [*p* = 0.030]; Table 1 and Figure 2a), and CAWS+statin group (3.5 ± 1.1 vs. 1.6 ± 0.5 [*p* = 0.020]; Table 1 and Figure 2a). Compared to the CAWS group, the CAWS+statin group showed a trend toward increased eNOS expression, but it was not significant (*p* = 0.066).

### 2.4. Expression of Phospho-Akt and Akt Protein in Whole Aortic Tissue

Akt expression was significantly higher in the CAWS group than in the control group (0.02 ± 0.02 vs. 0.37 ± 0.16 [*p* = 0.030]; Table 1 and Figure 2b), and significantly higher in the CAWS+statin group than in the CAWS group (0.37 ± 0.16 vs. 1.12 ± 0.33 [*p* = 0.020]; Table 1 and Figure 2b).

### 2.5. Expression of Cellular Senescence Markers in Whole Aortic Tissue

RNA was extracted from aortic tissue, and the expression levels of *p53*, *ICAM-1*, and vascular cell adhesion molecule 1 (*VCAM-1*) were measured. The expression of p53 was upregulated in the CAWS group compared with the control group, although the difference was not statistically significant (1.1 ± 0.1 vs. 2.9 ± 1.4 [*p* = 0.052]; Table 1 and Figure 3a). Furthermore, p53 expression was also significantly upregulated in the CAWS+statin group compared with the control group (1.1 ± 0.1 vs. 3.4 ± 1.4 [*p* = 0.037]; Table 1 and Figure 3a). However, there was no significant difference in p53 expression between the CAWS and CAWS+statin groups (*p* = 0.178).

*ICAM-1* mRNA expression was significantly upregulated in the CAWS group compared with the control group (1.2 ± 0.02 vs. 8.5 ± 6.0 [*p* = 0.004]; Table 1 and Figure 3b). Additionally, *ICAM-1* mRNA expression was significantly upregulated in the CAWS+statin group compared with the control group (1.2 ± 0.02 vs. 6.4 ± 3.2 [*p* = 0.012]; Table 1 and Figure 3b). Although *ICAM-1* expression in the CAWS+statin group was lower than in the CAWS group, the difference was not statistically significant (*p* = 0.61).

Analysis of *VCAM-*1 mRNA levels showed higher expression in the CAWS group compared with the control group (1.2 ± 0.2 vs. 11.9 ± 9.3 [*p* = 0.005]; Table 1 and Figure 3c). *VCAM-1* mRNA expression was also significantly higher in the CAWS+statin group than in the control group (1.2 ± 0.2 vs. 8.2 ± 4.8 [*p* = 0.023]; Table 1 and Figure 3c) and lower in the CAWS+statin group than in the CAWS group, although the difference was not statistically significant (*p* = 0.484).

Aortic tissue protein was used to evaluate the expression of phospho-Rb (Ser780), a marker of cellular senescence involved in the cell cycle. The expression of phospho-Rb was significantly higher in the CAWS group than in the control group (0.03 ± 0.02 vs. 0.31 ± 0.14 [*p* = 0.030]; Table 1 and Figure 3d), but significantly lower in the CAWS+statin group compared with the CAWS group (0.31 ± 0.14 vs. 0.08 ± 0.03 [*p* = 0.020]; Table 1 and Figure 3d).

### 2.6. Serum hs-CRP Levels in Each Group

Serum high-sensitivity C-reactive protein (hs-CRP) levels were significantly higher in the CAWS group than in the control group 10 weeks after CAWS administration (86.8 ± 15.3 pg/mL vs. 1127.2 ± 969.6 pg/mL [*p* = 0.012]; Table 1). However, hs-CRP levels in the CAWS and CAWS+statin groups did not differ significantly 10 weeks after CAWS administration (1127.2 ± 969.6 pg/mL vs. 840.4 ± 776.6 pg/mL [*p* = 0.403]; Table 1).

## 3. Discussion

CAWS vasculitis has previously been shown to promote atherosclerosis by stimulating the infiltration of inflammatory cells, including macrophages, into vascular tissue, while statins were shown to exert anti-inflammatory and anti-atherosclerotic effects by inhibiting inflammatory cell infiltration [8]. Low-density lipoprotein/very-low-density lipoprotein cholesterol levels were not sufficiently suppressed by statins, suggesting that statin-induced anti-atherosclerotic effects were influenced by pleiotropic effects rather than by the cholesterol-lowering effects of low-density lipoprotein [8,22,23]. However, the mechanisms through which CAWS promotes macrophage infiltration and statins inhibit the progression of inflammation and atherosclerosis remain unclear. Endothelial cell dysfunction may promote atherosclerosis, and we have considered the possibility that statins may improve endothelial cell dysfunction. However, vascular endothelial dysfunction has not been directly demonstrated. This present study revealed that chronic inflammation caused by CAWS vasculitis may promote vascular cell senescence, leading to impaired endothelial dysfunction, which in turn promotes macrophage invasion into vascular endothelial tissue. It was also speculated that statins maintain vascular endothelial cell homeostasis to reduce aortic atherosclerosis by enhancing eNOS expression.

Immunohistochemistry analyses showed that eNOS expression in the ascending aortic intima was significantly reduced following CAWS administration, and this effect was significantly enhanced by administering statins. On the other hand, Western blot analysis of proteins extracted from whole aortic tissue pointed to persistent downregulation of vascular endothelial cell function in the CAWS group compared with the control group, and administering a statin had no significant effect on increasing eNOS protein expression.

Both CAWS and CAWS+statin groups showed increased expression of eNOS in micro-neovessels and coronary artery aneurysms found in CAWS vasculitis-induced aneurysmal tissues (Appendix A). Protein samples were extracted from the entire aorta, including the aneurysmal tissue. Therefore, it might be difficult to show the effects of statins on eNOS via a comparison of the amount of eNOS protein, unlike the immunostaining findings. In contrast, Western blot analysis results suggested that treatment with a statin markedly increased Akt activity. Previous studies have shown that eNOS is activated via the PI3K/Akt/eNOS and Rho/ROCK pathways [17,24,25,26,27]. Moreover, statins cause phosphorylation of Akt in the PI3K/Akt/eNOS pathway [16,18,28,29,30]. In the current study, statin-induced increase in eNOS expression in the intimal epithelium was thought to mediate these pathways. In addition, Akt was activated in the CAWS group compared to the control group. Akt-related signaling is diverse [17], and Akt activity is increased by chronic inflammatory stimulation due to vasculitis [31,32,33]. In this study, serum hsCRP levels indicated persistent systemic inflammation in the CAWS and CAWS+statin groups. Since CAWS vasculitis causes persistent inflammation of blood vessels [8,34], we hypothesized that Akt activation in arterial tissues was due to chronic inflammation in the CAWS group. Impaired vascular endothelial cell function has been observed during vascular cell senescence, including decreased eNOS, ICAM-1, and VCAM-1 expression, as well as increased production of inflammatory cytokines [12,35,36,37,38,39]. These findings are indicative of early-stage atherosclerosis [38]. Cellular senescence is defined as a stable arrest of the cell cycle, with no growth of proliferatively competent normal cells [21,40,41]. The cellular senescence-induced DNA damage response arrests the cell cycle by activating Rb proteins via p53-p21 and p16-Rb pathways [19,42,43]. Since p53 forms multimeric complexes and their stability and activity vary [19,44,45,46], evaluating p53 activity using protein expression levels only is difficult. ICAM-1 and VCAM-1 are present as either soluble proteins, or are attached to vascular endothelial cells and blood cells [20,47,48]. *p53, ICAM-1,* and *VCAM-1* expression in vascular tissues was analyzed using qPCR. The results showed that CAWS vasculitis significantly increased vascular cellular senescence, whereas statins had no inhibitory effect on vascular cellular senescence (Figure 3). Serum hs-CRP levels in mice injected with CAWS were significantly higher than in mice in the control group (Table 1), with the CAWS group showing persistent inflammation in the aortic root and abdominal aorta [8,49]. The anti-inflammatory effect of statins alone was not enough to eliminate systemic inflammation, suggesting that vascular senescence might be more persistent in patients with KD than in healthy adults. Pleiotropic effects of statins are associated with apoptosis [17,50,51,52,53]. However, the effect of statins on cellular senescence needs to be evaluated from other perspectives as well.

This study had several limitations. First, this is an animal study that used mice with KD-like vasculitis. Proving the efficacy of statins against KD will require prospective clinical trials. Second, we found that statins improve vascular endothelial cell function by activating Akt and eNOS. However, additional in vitro experiments are required to confirm the underlying mechanisms and interactions. Finally, this study had a small sample size, potentially increasing the chances of experimental errors. Future studies will require larger sample sizes.

## 4. Materials and Methods

The study was carried out in accordance with the Guidelines for Animal Experiments of Nippon Medical School, the guidelines of the Law and Notification of the Government of Japan, and The ARRIVE Guidelines [54].

### 4.1. Animals

Five-week-old male C. KOR/StmSlc-ApoEshl mice were purchased from Sankyo Labo Service Co., Ltd. (Tokyo, Japan). Mice were maintained in a 12 h light/12 h dark cycle at 20–24 °C with 40%–70% humidity. A maximum of five mice were housed together, and the mice were checked for health and stress every day. To prevent sex-specific effects, only male mice were used. Water and food were provided ad libitum. Mice were fed a normal diet until they were 7 weeks old, at which point they were switched to a high-fat diet containing 0.15% cholesterol.

### 4.2. Preparation of CAWS and Statin

CAWS was prepared from the *C. albicans* strain NBRC1385, as previously described [55]. Briefly, 5 L of C-limiting medium was maintained in a glass incubator for two days at 27 °C and 400 rpm, with air being supplied at a rate of 5 L/min. An equal volume of ethanol was then added, and the mixture allowed to stand overnight, after which the precipitate was collected and dissolved in 250 mL of distilled water. Ethanol was added and the mixture allowed to stand overnight once again. The precipitate was subsequently collected, and dried with acetone to obtain CAWS.

The HMG-CoA inhibitor atorvastatin calcium hydrate (atorvastatin) was provided by Sankyo Ltd. (Tokyo, Japan). Atorvastatin was crushed and dissolved in 0.5% (*w*/*v*) methylcellulose 400 solution (WACO FUJIFILM CORPORATION, Tokyo, Japan) and sterilized. Statins were administered orally at the same time as the high-fat diet (from 7 weeks of age) at a dose of 10 mg/kg/day. The dose was determined based on the clinical dose given to humans. To eliminate errors in statin doses administered due to differences in individual food consumption, statins were administered directly using an oral sonde.

### 4.3. Experimental Procedures

Mice were divided into three groups (Figure 4) as follows. (1) Control group: 5-week-old ApoE− mice were intraperitoneally injected with phosphate-buffered saline (PBS) instead of CAWS. Additionally, instead of statins, 10 mg/kg of methylcellulose solution was given orally to the mice each day. (2) CAWS group: CAWS (4 mg/mouse) was intraperitoneally injected into 5-week-old ApoE− mice for five consecutive days, as described previously [49]. Instead of statins, the mice were given daily oral doses of methylcellulose solution. (3) CAWS+statin group: mice were given daily oral doses of statins beginning from 2 weeks after the CAWS administration (from 7 weeks of age) until the last day of the experiment (for 8 weeks).

Male mice reach sexual maturity at eight weeks of age. Fifteen weeks corresponds to adulthood in human years [56,57,58]. Since statins are not administered to infants in clinical practice, we designed a model in which statins are administered in early adulthood. All mice were fed a high-fat diet containing 0.15% cholesterol for two weeks. Five mice from each group were euthanized 10 weeks after CAWS injection (at 15 weeks of age), and their hearts and aorta, en face, were extracted for analysis. 

### 4.4. Assessment of Horizontal Transections of Ascending Aorta

The mice were anesthetized, and the aortas excised from the aortic arch to the iliac bifurcation. Frozen sections of the ascending aorta were cut from tissue samples embedded in Optimal Cutting Temperature compound, and were then stained for immunohistochemistry analysis. Briefly, the samples were incubated with anti-eNOS antibody (1/100, Abcam; ab5589, UK) at 37 °C for 120 min. The sections were then treated with secondary antibodies and developed using HRP-conjugated DAB substrate (Abcam; ab236446, UK). Horizontal sections of eNOS-stained samples were quantified and analyzed using the Hybrid Cell Count System (KEYENCE) and KEYENCE BZX analyzer (Osaka, Japan). Each imaged aorta was divided into 12 sections (Figure 1b), and the degree of eNOS expression scored as follows: score 0, no staining; score 1, partially stained; score 2, partially continuously stained; score 3, totally continuously stained [59]. The images were analyzed by three trained observers who were blinded to the treatment of the mice.

### 4.5. Western Blot Analysis of Proteins Extracted from Aortic Tissue

Aortic tissue from the ascending aorta to the bifurcation collected at sacrifice was shredded and proteins harvested as previously described [60,61]. Briefly, to harvest proteins, 0.5 mL RIPA buffer containing protease inhibitor and no Triton X-100 was added to the tissue, and the tissue homogenized for 5 min at the highest frequency using Tissue Lyser II (Qiagen, Venlo, Nederland). The supernatant was isolated by centrifugation at 12,000 rpm and 4 °C for 10 min. The protein concentration in the supernatant was measured and Western blotting performed as follows: the collected proteins were boiled in SDS-PAGE sample loading buffer, separated by SDS-PAGE, and transferred to PVDF membranes (Millipore). The concentration of each sample was adjusted to 14 μg per well and electrophoresis was performed. Membranes were blocked with 10% nonfat dried milk in TBST buffer (20 mM Tris-HCl (7.5), 140 mM NaCl, and 0.1% Tween-20), probed with primary antibodies, incubated with horseradish peroxidase-conjugated mouse or rabbit immunoglobulin G (GE Healthcare, England), and then visualized using Chemi-Lumi ONE Ultra (Nacalai Tesque). Protein bands were analyzed using LAS-4000 mini image analyzer (Fujifilm, Japan) and the intensity of each band was quantified using ImageJ software (ver 1.5.3). For quantification, the intensity of the GAPDH band was used to normalize each protein signal.

### 4.6. Antibodies and Materials

Anti-eNOS polyclonal antibodies (1:1000, ab5589; Abcam, Cambridge, UK), anti-phospho-Akt (Thr308) polyclonal antibodies (1:1000, #9275; Cell Signaling Technology, Danvers,MA, USA), anti-Akt polyclonal antibodies (1:1000, #9272; Cell Signaling Technology, Danvers,MA, USA), anti-phospho-Rb (Ser780) monoclonal antibodies (1:1000, M045-3; MBL, Japan), anti-GAPDH monoclonal antibodies (1:1000, sc-32233; Santa Cruz Biotechnology, Dallas, TX, USA), and anti-ICAM-1 (M-19) monoclonal antibodies (1:1000, sc-1511; Santa Cruz Biotechnology, Dallas, TX, USA) were used for Western blot analysis.

### 4.7. RNA Isolation and Quantitative PCR

Genomic material was extracted from shredded aortic tissue, and RNA isolated using the NucleoSpin RNA kit (Takara Bio, San Jose, CA, USA) according to the manufacturer’s instructions. Equal amounts of RNA (1.0 µg) were reverse-transcribed using PrimeScript II 1st strand cDNA Synthesis Kit (Takara Bio). The resulting cDNA was subjected to quantitative PCR (qPCR) analysis using TaqMan Gene Expression Assay Master mix and qPCR probes (Thermo Fisher Scientific, Waltham, MA, USA), and Step One Plus Real-Time PCR System (Thermo Fisher Scientific). The TaqMan gene expression assays used to analyze the mouse samples were p53 (Mm01731287_m1), ICAM-1 (Mm01175876_g1), and VCAM-1 (Mm00449197_m1). Each amplification reaction was performed in triplicate, and the mean and standard deviation of three threshold cycles were used to calculate the quantity of transcripts in the sample (StepOne software v2.2.2; Thermo Fisher Scientific). The quantity of mRNA was expressed in arbitrary units as the ratio of the sample quantity to the calibrator, or the mean values of control samples. All values were normalized to the endogenous control, mouse GAPDH (Mm99999915_g1).

### 4.8. Serological Evaluation

Serum samples were stored at −20 °C until analysis. Sandwich enzyme-linked immunosorbent assay was used to detect the plasma levels of hs-CRP (MyBioSource, San Diego, CA, USA; hs-CRP elisa kit, MBS262829) following the manufacturer’s instructions.

### 4.9. Statistical Analysis

Statistical analyses were performed using JMP statistical software version 12 (SAS Institute Inc., Cary, NC, USA). The Kruskal–Wallis test was used to analyze the significance of statistical differences between the groups. When significance was detected, the Wilcoxon test was used as a post hoc test to compare the values between groups. Statistical data are expressed as median (upper and lower quartiles) or mean ± standard deviation. Statistical significance was set at *p* < 0.05.

## 5. Conclusions

CAWS vasculitis, a possible KD model, promoted vascular senescence in the aorta and decreased vascular endothelial cell function. Statins improved vascular endothelial cell function by activating Akt, and inducing eNOS expression in vascular endothelial cells. Statins might suppress the infiltration of inflammatory cells, including macrophages, by restoring vascular endothelial cell function impaired by KD-like vasculitis. Thus, statin therapy might suppress vasculitis-induced atherosclerosis that develops in early adulthood. The results of this study potentially expand the indications for statin therapy in patients with KD.

## Figures and Tables

**Figure 1 ijms-23-16108-f001:**
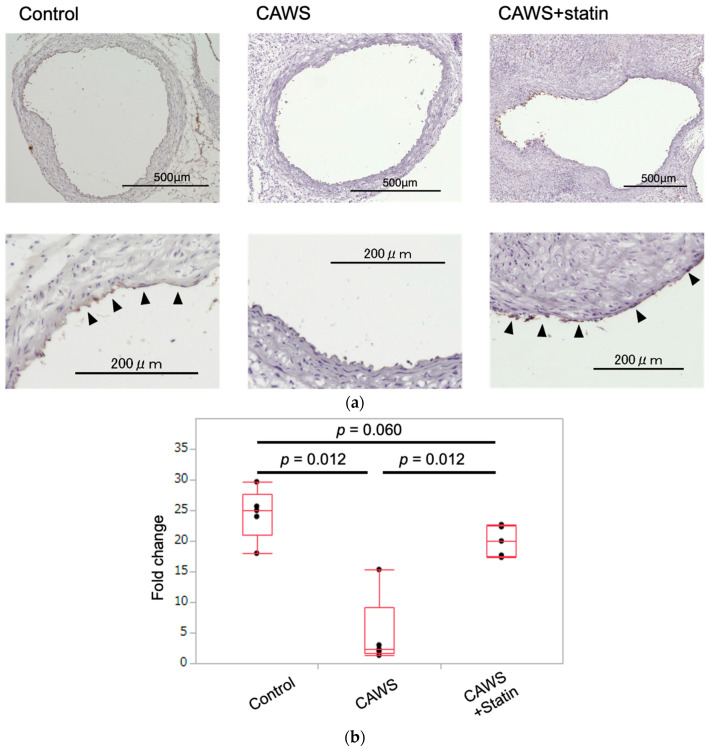
Endothelial nitric oxide synthase (eNOS) staining and scoring of the strength of eNOS expression in ascending aorta. (**a**) Frozen transverse sections of the ascending aorta were immuno-stained to detect eNOS. Near-continuous eNOS expression (black arrowheads) was observed in the control group but was suppressed in the CAWS group. Partial eNOS expression was observed in the CAWS+statin group (upper bar = 500 μm; lower bar = 200 μm). (**b**) The aorta was divided into 12 sections and the degree of eNOS expression in each region scored on a 36-point scale from 0 to 3. eNOS expression was significantly suppressed in the CAWS group compared with the control group and was upregulated in the CAWS+statin group compared with the CAWS group.

**Figure 2 ijms-23-16108-f002:**
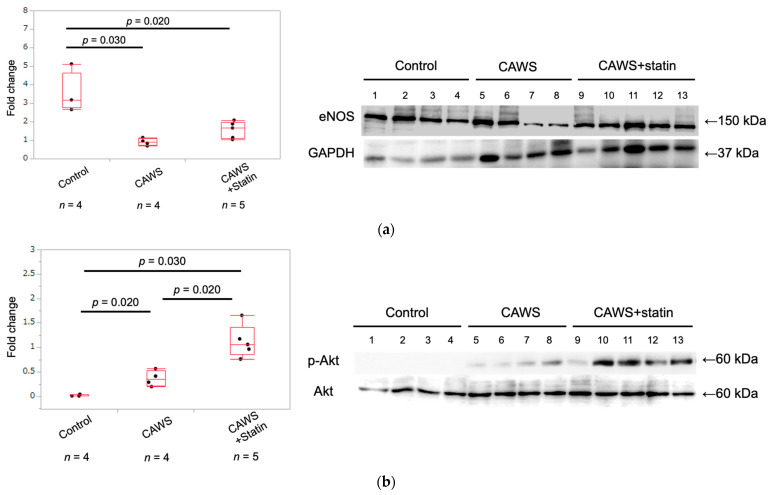
Expression of eNOS, phospho-Akt, and Akt proteins in aortic tissue. (**a**) The intensity of eNOS protein expression was normalized to that of glyceraldehyde 3-phosphate dehydrogenase (GAPDH). eNOS expression was significantly higher in the control group than in the CAWS and CAWS+statin groups. The CAWS+statin group had higher but non-significant eNOS expression compared with the CAWS group. (**b**) Akt expression was significantly higher in the CAWS and CAWS+statin groups compared with the control group. Comparison of the CAWS and CAWS+statin group showed higher Akt expression in the CAWS+statin group.

**Figure 3 ijms-23-16108-f003:**
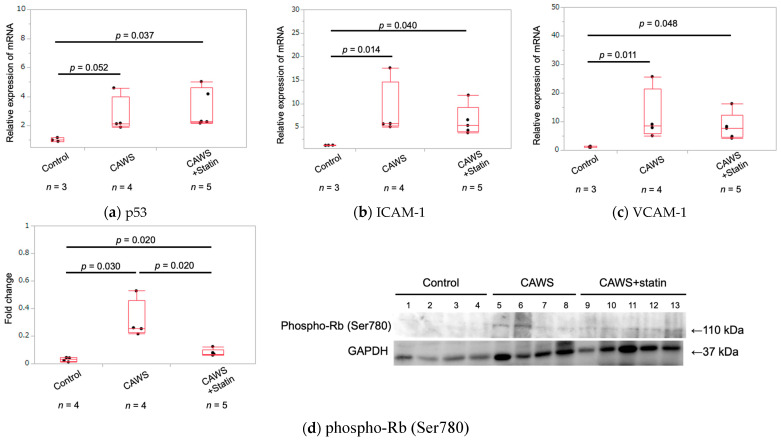
Analysis of mRNA and protein expression in aortic tissue. (**a**–**c**) The expression of *p53, ICAM-1,* and *VCAM-1* was evaluated. Evidence of progressive cellular senescence in the vessels was observed following CAWS administration. Cellular senescence was not significantly inhibited following treatment with statin. (**d**) Analysis of phospho-Rb (Ser780) protein expression in aortic tissue. Phospho-Rb (Ser780) expression was higher in the CAWS and CAWS+statin groups than in the control group. These findings indicate that CAWS vasculitis causes progressive cellular senescence in vascular tissues, and treatment with statin suppresses this expression.

**Figure 4 ijms-23-16108-f004:**
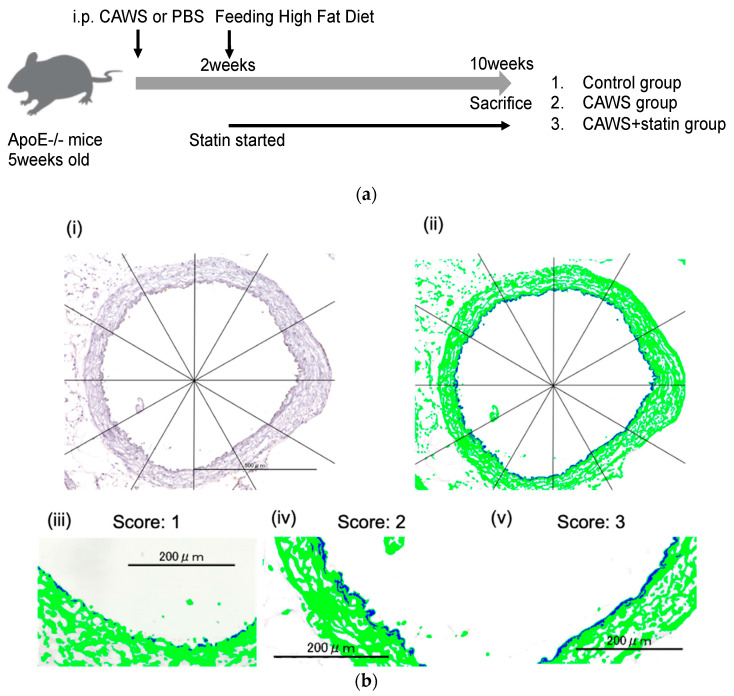
Experimental procedure. (**a**) 4 mg of CAWS was intraperitoneally injected into each 5-week-old ApoE− mouse for five consecutive days to induce Kawasaki disease-like vasculitis. Two weeks after administering CAWS, the mice were switched to a high-fat diet to promote atherosclerosis. Daily statin doses (10 mg/kg/day) were administered beginning two weeks after CAWS administration and until the end of the experiment. Mice were sacrificed and assessed 10 weeks after CAWS administration. This schematic illustration was modified from our previous published study [8]. (**b**) Assessment of the horizontal transection area of the ascending aorta (bar, 200 μm). eNOS-stained horizontal section samples (i) were imaged using (KEYENCE) and KEYENCE BZX analyzer (Osaka, Japan). The immunostained regions of the aorta were extracted using KEYENCE BZX analyzer (ii). The aorta was divided into 12 sections and the degree of eNOS expression in each region defined based on a scoring system: score 0, no staining; score 1, partially stained (iii); score 2, partially continuously stained (iv); score 3, totally continuously stained (v). Each specimen was scored on a 36-point scale.

**Table 1 ijms-23-16108-t001:** Summary of the data collected 10 weeks after *Candida albicans* water-soluble fraction (CAWS) administration.

10 Weeks after CAWS Administration						
	Control		CAWS		CAWS+Statin	
Body weight (g)	32.6 ± 2.9		29.1 ± 2.7		25.6 ± 2.5 *	
	(*n* = 5)		(*n* = 5)		(*n* = 5)	
Spleen weight (g)	0.14 ± 0.01		0.29 ± 0.05 *		0.22 ± 0.04 **	
	(*n* = 5)		(*n* = 5)		(*n* = 5)	
Spleen/Body weight ratio(%)	0.44 ± 0.04		0.98 ± 0.11 *		0.87 ± 0.11 *	
	(*n* = 5)		(*n* = 5)		(*n* = 5)	
**Serological Evaluation**						
hs-CRP (pg/mL)	86.8 ± 15.3		1127.2 ± 969.6 *		840.4 ± 776.6 *	
	(*n* = 5)		(*n* = 5)		(*n* = 5)	
**Immunohistochemical staining**						
eNOS scoring in ascending aorta	25.0	[27.7, 21]	2.3	[9.2, 1.7] *	20.0	[22.5, 17.5] **
	(*n* = 5)		(*n* = 5)		(*n* = 5)	
**Western blot analysis for Aortic tissue**						
Expression of eNOS protein	3.5 ± 1.1		0.9 ± 0.2 *		1.6 ± 0.5 *	
	(*n* = 4)		(*n* = 4)		(*n* = 5)	
Phospho-Akt/Akt ratio	0.02 ± 0.01		0.4 ± 0.16 *		1.12 ± 0.33 *^,^ **	
	(*n* = 4)		(*n* = 4)		(*n* = 5)	
Expression of phosoho-Rb(Ser780)	0.03 ± 0.02		0.31 ± 0.14 *		0.08 ± 0.03 *^,^ **	
	(*n* = 4)		(*n* = 4)		(*n* = 5)	
**Quantitative PCR analysis for Aortic tissue**						
p53	1.1 ± 0.1		2.9 ± 1.4 *		3.4 ± 1.4 *	
	(*n* = 3)		(*n* = 4)		(*n* = 5)	
ICAM-1	1.2 ± 0.02		8.5 ± 6.0 *		1.4 ± 3.2 *	
	(*n* = 3)		(*n* = 4)		(*n* = 5)	
VCAM-1	1.2 ± 0.2		11.9 ± 9.4 *		8.2 ± 4.8 *	
	(*n* = 3)		(*n* = 4)		(*n* = 5)	

* *p* < 0.05 compared to Control; ** *p* < 0.05 compared to CAWS.

## Data Availability

The datasets generated and/or analyzed during the current study are available from the corresponding author on reasonable request.

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
