# Peer review of "Statins Show Anti-Atherosclerotic Effects by Improving Endothelial Cell Function in a Kawasaki Disease-like Vasculitis Mouse Model"

_ijms, 2022, doi:10.3390/ijms232416108_

Round 1
Reviewer 1 Report
Dear Authors,
I find your manuscript extremely interesting, very well conceived and conducted, and its findings and subsequent potential applications can really improve KD patients outcomes.

Reviewer 2 Report
This paper is related to "Motoji Y, et al. Biomedicines. 2022". This paper is well written, and the text is clear and easy to read. However, compared to the previous work, the conclusion is similar in supporting the efficacy of statins for Kawasaki disease. Thus, this content is of little significance. Comments and questions are shown below.
In the introduction, the authors may add why they focused on eNOS in this study.
In the Figure 1 legend, arrowheads would be appropriate instead of arrows.
I'm wondering if 3 to 5 subjects analyzed is appropriate due to the data variability and subtle significant differences across the figures. Isn't it possible to reanalyze with a larger number of subjects?
Reviewer 3 Report
This study uses an atherosclerosis prone rodent to study the effects of statins on endothelial function in a model of Kawasaki disease.
Could the authors please provide lipid data for the 3 rodent groups - before and after use of statin. This would be of interest since atherosclerosis is in part a lipid disorder.
Is use of an atherosclerosis prone mouse the correct type of animal to study KD? After all KD is generally in younger people where atherosclerosis is not an issue.
Round 2
Reviewer 2 Report
Please consider increasing the number of samples in the future.